# Retrieval of Crude Protein in Perennial Ryegrass Using Spectral Data at the Canopy Level

**Gustavo Togeiro de Alckmin** [1,2,]*[ ], **Arko Lucieer** [1][ ], **Gerbert Roerink** [3], **Richard Rawnsley** [4][ ], **Idse Hoving** [5] **and Lammert Kooistra** [2][ ]

[1] School of Technology, Environments and Design, University of Tasmania-Discipline of Geography and Spatial Sciences, Hobart, TAS 7005, Australia; arko.lucieer@utas.edu.au
[2] Laboratory of Geo-Information Science and Remote Sensing, Wageningen University, Droevendaalsesteeg 3, 6708 PB Wageningen, The Netherlands; Lammert.kooistra@wur.nl
[3] Wageningen Environmental Research-Earth Informatics, Droevendaalsesteeg 3, 6708 PB Wageningen, The Netherlands; gerbert.roerink@wur.nl
[4] Tasmanian Institute of Agriculture-Centre for Dairy, Grains and Grazing, 16-20 Mooreville Rd, Burnie, TAS 7320, Australia; Richard.Rawnsley@fonterra.com
[5] Wageningen Livestock Research-Livestock and Environment, De Elst 1, 6700 AH Wageningen, The Netherlands; Idse.Hoving@wur.nl
* Correspondence: gustavo.alckmin@utas.edu.au

**Abstract:** Crude protein estimation is an important parameter for perennial ryegrass (*Lolium perenne*) management. This study aims to establish an effective and affordable approach for a non-destructive, near-real-time crude protein retrieval based solely on top-of-canopy reflectance. The study contrasts different spectral ranges while selecting a minimal number of bands and analyzing achievable accuracies for crude protein expressed as a dry matter fraction or on a weight-per-area basis. In addition, the model's prediction performance in known and new locations is compared. This data collection comprised 266 full-range (350–2500 nm) proximal spectral measurements and corresponding ground truth observations in Australia and the Netherlands from May to November 2018. An exhaustive-search (based on a genetic algorithm) successfully selected band subsets within different regions and across the full spectral range, minimizing both the number of bands and an error metric. For field conditions, our results indicate that the best approach for crude protein estimation relies on the use of the visible to near-infrared range (400–1100 nm). Within this range, eleven sparse broad bands (of 10 nm bandwidth) provide performance better than or equivalent to those of previous studies that used a higher number of bands and narrower bandwidths. Additionally, when using top-of-canopy reflectance, our results demonstrate that the highest accuracy is achievable when estimating crude protein on its weight-per-area basis (RMSEP 80 kg.ha$^{-1}$). These models can be employed in new unseen locations, resulting in a minor decrease in accuracy (RMSEP 85.5 kg.ha$^{-1}$). Crude protein as a dry matter fraction presents a bottom-line accuracy (RMSEP) ranging from 2.5–3.0 percent dry matter in optimal models (requiring ten bands). However, these models display a low explanatory ability for the observed variability ($R^2 > 0.5$), rendering them only suitable for qualitative grading.

**Keywords:** perennial ryegrass; hyperspectral; machine learning; crude protein; partial least squares; feature selection; variable importance

---

## 1. Introduction

Timely assessment of feed quality parameters is necessary for optimal management of pasture-based dairy (livestock) systems. Such monitoring can instruct well-informed decisions in grazing (feeding) management according to whichever nutritional goals are established [1]. Currently, these parameters are usually estimated through laboratory analysis, involving time-consuming sampling procedures and complex logistical operations, and are subject to the availability of a service provider [2,3]. In practice, most pasture managers often rely on proxies such as seasonal patterns, plant maturity, morphology, or ecophysiological relations [4], which may not provide the accuracy or precision required for optimal pasture management [5]. Within perennial ryegrass's (*Lolium perenne*) macronutrients, *crude protein* (CP) usually displays the largest variability [3], directly responding to soil–plant–animal interactions and seasonal patterns, whilst being a key component of optimal ruminal activity and, consequently, intake and digestibility [6].

Feed quality assessment is ultimately based on wet-chemistry methods, which are usually expensive, complex, and time consuming. Inversely, *near-infrared spectroscopy* (NIRS) is an established spectral-based laboratory technique that is able to reduce costs and increase throughput while providing estimates equivalent to such reference methods [2,7]. Ideally, employing non-destructive spectral analysis in field conditions would be advantageous to farmers, as accurate real-time information allows precise management of a herd's diet and understanding of short- and long-term effects of different strategies for grazing (e.g., grazing interval and pressure) and pasture management (e.g., timing and application rates of inputs).

To this end, *remote sensing* (RS) has been employed in several studies [8–11], where different spatial, spectral, and data acquisition scales (e.g., proximal, low-level flight, airborne, and satellite) were examined. None, however, have found the same level of accuracy for crude protein retrieval as benchtop NIRS, demonstrating the challenging nature of the straightforward translation of laboratory methods to field conditions. In contrast to processed laboratory samples (i.e., dried and ground), spectral field measurements are impaired by non-homogeneous biophysical attributes of targets (e.g., moisture, *leaf area index* (LAI), *leaf angle distribution* (LAD)) [12], in which the underlying fundamentals of spectroscopy (Beer–Lamberts law) [11,13] are not entirely observed.

In outdoor environments, measurements and analyses are increasingly complex given the dynamic and reactive nature of plant–light interactions and nitrogen mobility through the plant [14,15]. More importantly, field spectroscopy is an ill-posed problem as per Hadamards's definition [16]: an observed spectral response can be resolved by multiple combinations of parameters (e.g., pigment concentration, optical thickness), resulting in ambiguous estimations of the target's true constituents [12,17]. Consequently, in the absence of prior biophysical information, plant-tissue biochemical characterization based solely on *top-of-canopy* (TOC) reflectance is inherently poorly determined.

To cope with such limitations, two approaches have been extensively utilized [18–22]: (i) the use of spectral shapes and transformations (e.g., derivatives and continuum removal) and (ii) the estimation of the canopy's biochemical attributes in conjunction with biophysical properties (e.g., biomass per unit area). This second approach has led to the retrieval of biochemical estimates as mass per unit area (i.e., kg.ha$^{-1}$) rather than as a *dry matter* (DM) fraction (i.e., %DM) [22,23].

While spectral transformations (i) are used as a tool to disentangle overlapping absorption features and ameliorate light scattering, in practice, this potential has not been confirmed in uncontrolled environments when employed as a preprocessing tool to improve estimations [18,24]. Additionally, in a farm scenario, their adoption would be limited by the necessary number of contiguous bands, narrow bandwidth, radiometric sensitivity, and high signal-to-noise ratio necessary for their accurate computation. In contrast to transformations, reflectance measurements can be executed in sparse bands, with broader bandwidths and coarser radiometric resolution. Hence, a plausible alternative is the (ii) estimation of biochemical attributes in terms of mass per area through reflectance measurements [25].

Despite not achieving the same levels of accuracy as NIRS, from a pasture management perspective, the absence of extremely accurate estimates (i.e., within less than 1% of the reference method) is offset by continuous non-destructive estimations of acceptable precision, provided that these are made available at a low cost and without geographical or temporal restrictions. These requirements can be met through the development of a spectral model with a small number of bands (i.e., feature selection), validated under representative sampling conditions (i.e., different locations, dates, or seasons). Such conditions are essential to instruct the design of an effective, low-cost sensor for CP estimation.

Notably, hyperspectral data present a high degree of multicollinearity (i.e., redundancy of explanatory variables), which is usually handled through multivariate techniques, such as *partial least squares regression* (PLSR). However, a common misconception is the assumption that the full spectral range should yield better results than the causal or informative part of the spectrum [26], allowing for a feature selection routine without a corresponding loss in model prediction accuracy [27]. However, due to overlapping absorption features and multicollinearity, an upfront and deliberate choice of optimal spectral ranges or band subsets (i.e., explanatory variables) is not trivial. Multicollinearity also renders model comparison and performance analysis difficult,as different subsets of bands can be employed with equivalent model performance.

Approaching these issues, Kawamura et al. [19] provides an insight into feature selection for *crude protein yield* ($CP_m$, kg CP.ha$^{-1}$) and achievable accuracies by employing both iterative stepwise elimination and *genetic algorithm* (GA) routines, reporting the best results in the latter ($R^2 = 0.79$ and 0.85, respectively). The authors also applied the same routines to estimate crude protein as a *dry matter fraction* (%CP) and, as expected, reported lower accuracies than $CP_m$ ($R^2 = 0.30$ and 0.27, respectively).

To a large extent, the cost requisite can be tackled by a restriction of spectral regions employed in models and, consequently, sensors. *Visible and near-infrared* (VIS-NIR, 400–1100 nm) instruments, mostly *Silicon*-based (SI) semiconductors, mass-produced for off-the-shelf consumer cameras, are less expensive than *shortwave infrared* (SWIR, 1100–2500 nm) instruments, which are mostly based on *indium gallium arsenide* (InGaAs) or *lead sulfide* (PbS). Discussing this issue, Starks et al. [24] reported that CP (%CP and $CP_m$) could be best estimated using the VIS-NIR rather than the SWIR portion of the spectrum.

By joining these methods [19,24], this study aims to assess model performances coupled with routines of feature selection within different spectral regions (i.e., VIS-NIR or SWIR) or the *full spectrum* (FS) range. This analysis can inform both the minimal necessary spectral measurements for crude protein retrieval as well as the design of a *multispectral* (MS) sensor (either a point-measurement or imaging system) that is less complex and costly than hyperspectral instruments employed in laboratory analysis.

In summary, this research aims to predict perennial ryegrass $CP_m$ and %CP after routines of feature selection (i.e., genetic algorithm) by employing either VIS-NIR, SWIR, or the full-range spectrum as predictors. Spectral transformation (e.g., derivatives or continuum-removed features) are not to be employed, as these require either contiguous or specific bands and low radiometric error, which may prevent the transferability of findings to a rural scenario. Finally, the analysis of the data should indicate a necessary minimal number of spectral bands for the design of a multispectral sensor at a low cost with an acceptable accuracy, which can be integrated into an autonomous system (unmanned aerial or ground vehicle) for a higher level of automation in data collection.

## 2. Methods

The experimental units consisted of rain-fed perennial ryegrass plots managed under different *nitrogen* (N) fertilization regimes and mowing intervals, generating a quantitative and qualitative gradient. The combination of these factors (i.e., nitrogen rate and regrowth interval) was randomized across each pseudo-replicate (Figure 1A,B). Nitrogen (supplied as urea) was broadcast uniformly across each plot. For all plots, the residual mowing height was approximately 50 mm (Figure 1II).

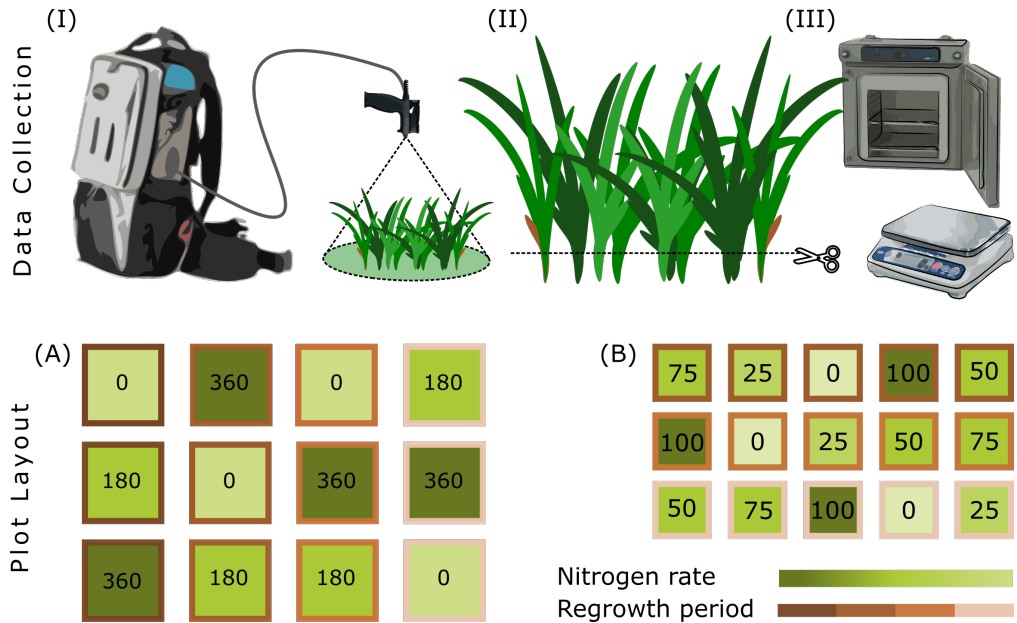

**Figure 1.** Data collection protocol (**I–III**) and plot layout (basic pseudo-replicate, (**A,B**)). Top: (**I**) Spectral measurements, (**II**) mechanical defoliation, and (**III**) drying and weighing. Bottom: basic Dutch (**A**) and Australian (**B**) plot layouts. Borderline colors indicate regrowth period, and hues of green (darker = higher rates) indicate nitrogen levels (rates are also indicated within each plot).

Four experimental sites were available: Elliot, Goutum, Vredepeel, and Zegveld. These sites were located at : Elliot (41°4′57.7″S, 145°46′22.0″E), Goutum (53°10′38.8″N, 5°46′27.0″E), Vredepeel (51°32′57.9″N, 5°51′54.0″E), and Zegveld (52°8′32.9″N, 4°50′23.5″E). While the first is located in Australia (Tasmania), the remaining sites are in the Netherlands. The data collection period started in early May and finished in late November 2018 (Table 1).

**Table 1.** Metadata associated with the data collection per location.

| Location | Date 1 | Date 2 | Date 3 | Instrument | Soil Type | N Rates * | Regrowth ** |
|---|---|---|---|---|---|---|---|
| Elliot (AU) | 11 November | 17 November | 24 November | FieldSpec 4 | Clay | 50–75–100 | 3 |
| Goutum (NL) | 15 May | 28 June | 05 October | FieldSpec 3 | Clay | 180–360 | 4 |
| Vredepeel(NL) | 11 June | 03 July | 27 September | FieldSpec 3 | Sandy | 180–360 | 4 |
| Zegveld (NL) | 10 May | 19 June | 10 October | FieldSpec 3 | Peat | 180–360 | 4 |

\* kg N.ha$^{-1}$; ** Weeks/Mowing Cycles.

## 2.1. Experimental Setup

The Dutch experimental sites (Figure 1A) contained 24 plots per location. In each, the layout was a factorial combination of three N fertilization levels (0, 180, and 360 kg N.ha$^{-1}$ per year), four mowing intervals (in a cycle of four weeks, when all plots were mown) and two pseudo-replicates of this combination, as presented in Hoving et al. [28]. Each site is located within a different soil type (either clay, sandy, or peat). Data collection spanned from May to October 2018. From the second half of July till September, data collection was interrupted due to a prolonged heatwave and drought, which constrained plant growth.

The Australian site (Figure 1B) consisted of 30 plots, managed under five different N levels (0, 25, 50, 75, and 100 kg N.ha$^{-1}$) applied on 20th October at a clay soil location. A set of 10 distinct plots were mowed on 11th and 17th November, resulting in three different regrowth periods. Data collection took place on the 11th, 17th, and 24th of November, 2018.

To provide evidence that these models are not restricted by location or season, data were collected in four locations (one in Australia and three in the Netherlands), spanning from spring to autumn

(i.e., the most important growth period) during early May to late November 2018. Nitrogen levels were set based on previous research work and in accordance with appropriate ranges of fertilizer levels commonly applied in each production system [6,28,29]. Dates were chosen to portray the most important growth periods (i.e., spring and autumn) for cool-season grasses. Only in Australia were all data collected in spring (November), as the instrument was available for a limited period of time. However, Australian biomass and %CP ranges were comparable to those of the locations in the Netherlands and were analyzed within the *Ground Truth Analysis*.

## 2.2. Data Collection

An identical data collection protocol was performed across sites and dates (Figure 1I–III ). Spectral measurements, ranging from 350 to 2500 nm, were taken in clear-sky periods at around solar noon $\pm 2$ h. Total time employed for spectral data collection ranged from 45 min to one hour. An ASD FieldSpec® 3 and FieldSpec® 4 (Malvern PanAnalytical–Boulder, CO, USA) were employed in the Netherlands and Australia, respectively. Both instruments had no attached fore optics (i.e., bare fiber; field of view: 25°) and were configured using the following setup (i.e., number of scans): 60 for white reference (Spectralon$^{TM}$; Labsphere–, North Sutton, NH, USA), 60 for dark current, 30 per measurement, and five measurements of each target (Figure 1I). The average of these five measurements was utilized for analysis purposes as the target's reflectance value. Measurements were taken at approximately one meter high (Figure 1I), resulting in a circular footprint of approximately 0.44 m diameter.

After the measurements of each plot were taken, a reference scan of the white reference was recorded for quality assurance. This procedure aimed to create a pseudo dual field of view by which the target's reflectance could be further corrected in case of changing ambient light (e.g., cloud formation) [30]. The instrument was recalibrated whenever the white reference measurement deviated from a straight line centered at one or a maximum time limit of seven minutes between recalibrations was reached, whichever occurred first.

The footprint was mechanically defoliated to a specific residual height (i.e., 50 mm) and stored in perforated plastic oven bags (Figure 1II). This residual height corresponds to the best-practice perennial ryegrass management [1] as well as height strata that have little to no light interaction [31]. The harvested material was immediately refrigerated and transported from the experimental sites to a forced-air oven, where it was dried for 48 h at 65 °C and weighed (Figure 1III).

The samples were then taken to a third-party feed quality laboratory (EuroFins Agro, Wageningen, NL and FeedTest, Werribee, AU), where %CP was estimated using proprietary NIRS methods. Crude protein fraction estimates are expected to be within less than 1.0% *Root Mean Square Error* (RMSE) [32] of the wet-chemistry method results.

Observations with sample weights of above 3500 kg DM.ha$^{-1}$ were excluded from the data set as these are not biomass levels to be achieved within the best pasture management practices [1] (target pre- and post-grazing biomass of 2400 and 1500 kg DM.ha$^{-1}$, respectively) and, thus, should not be considered within the representative conditions under which these models should operate.

## 2.3. Data Analysis

Data analysis was performed in RStudio/R (R Core Team [33], versions 1.2.5 and 4.0.2, respectively). The main necessary packages for the analysis, besides the base and dependencies packages, are *rsample* [34], *desirability* [35], *plsVarSel* [36], and *caret* [37]. For reproducibility purposes, the data analysis operations are introduced in the corresponding *package::function* format (italics typeface and accompanied by the double colon operator, i.e., the scope resolution operator).

### 2.3.1. Ground Truth Analysis

**Comparability between locations**: Measurements underwent a series of *post hoc* tests (i.e., analysis of variance) to ensure that data collected between locations were comparable,

testing whether the average values ($\mu$) and variances ($\sigma^2$) were significantly different ($H_0$: $\mu_i = \mu_j$ at $\alpha = 0.05$). Initially, the Bartlett Test of Homogeneity of Variances (*stats::batlett.test*) was performed, followed by Kruskal–Wallis (*stats::kruskal.test*) and Dunn's Multiple Comparisons (*FSA::dunnTest*). The Kruskal-Wallis (KW) test is a non-parametric method for testing whether samples originate from the same distribution. Pearson correlation ($\rho$ and its *p*-value, *stats::cor.test*) between biochemical and biophysical attributes was also computed to identify spurious relationships between spectral-based models and the main biophysical property (i.e., biomass). This should indicate whether the $CP_m$ models were estimating simply biomass or, indeed, the attribute of interest (i.e., $CP_m$).

**Missing data and outlier detection**: Two samples had missing %CP values, as these did not reach the minimum sample weight required for testing. For these, %CP values were imputed (*missMDA::imputePCA*) using the methods described in Josse and Husson [38], using spectral data and metadata (e.g., collection date, location) as predictors. Crude protein yield ($CP_m$) was calculated as the product of %CP per the sample's dry matter weight. Outlier detection was performed (*enpls::enpls.od*) as per the workflow described in Cao et al. [39]. The top ten observations that accounted for the highest error standard deviation or error mean were excluded from the analysis data set. This process is equivalent to other preliminary analyses, as the inspection of the inner relations of partial least squares (PLS) (i.e., TU plot) where the scores of both the predictor ($T$) and response variables ($U$) are plotted in order to identify outliers or the analysis of Q residuals and Hotelling Distance [40].

### 2.3.2. Spectral Analysis

**A. Pre-Processing**: Raw reflectance measurements were exported to ASCII format using ViewSpecPro$^{TM}$ and merged (*dplyr::right_join*) with the sample's metadata and feed quality information. The raw target reflectances were summarized as the average of the five measurements, and a data frame with a single reflectance value per measurement was created. This data frame was transformed into a Speclib object (*hsdar*, Lehnert et al. [41]); spectra were corrected at 1000 and 1800 nm (*prospectr::spliceCorrection*—FieldSpec sensor artifacts), followed by a smoothing process using a Savitzky–Golay filter (*hsdar::noiseFiltering*, window-size = 19 nm) and resampling (*hsdar::spectralResampling*) to 10 nm (*full width half maximum—FWHM*) contiguous bands using a gaussian distribution as a response function. The Savitzky–Golay filter was set in such a way to remove high-frequency noise and to prevent over-smoothing; the window size was similar to the value (i.e., 15 nm) employed in Kawamura et al. [19]. Bands associated with water vapor absorption features and noisy areas were identified and excluded (i.e., 350 to 400, 1325 to 1475, 1775 to 2000, and 2400 to 2500 nm) from the final data frame (Figure 2: "Pre-processing"). For visualization and reference purposes, an exploratory analysis was performed by grouping and averaging the spectra according to quartiles (either $CP_m$ or %CP).

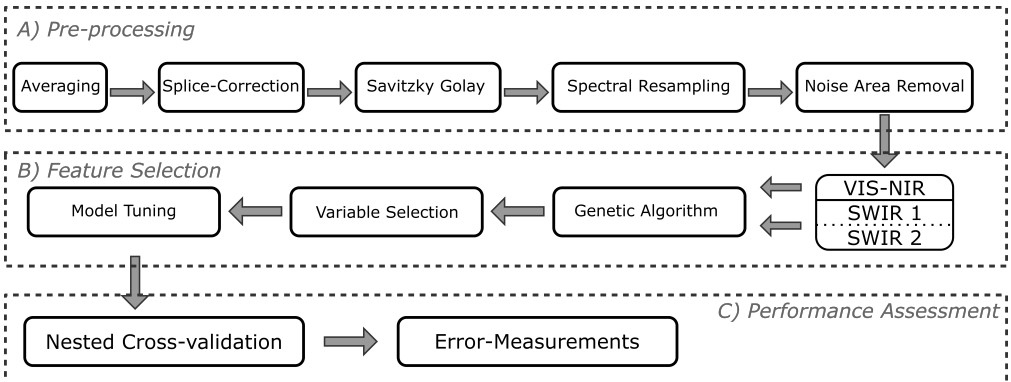

**Figure 2.** Data analysis workflow. The workflow is divided into (**A**) pre-processing, (**B**) feature selection, and (**C**) performance assessment. Within the feature selection, three sets (visible and near-infrared (VIS-NIR), shortwave infrared (SWIR), and full spectrum (FS)) were analyzed in parallel for further performance assessments.

**B. Feature (Band) Selection**: The workflow for feature selection is described in Kuhn and Johnson [42] ([ch. 19]). A genetic algorithm (GA) routine for feature (i.e., spectral bands) selection (*caret::gafs*) coupled with PLSR (*pls::plsr*, Mevik and Wehrens [43]) was carried out using the full spectrum, VIS-NIR (400–1100 nm bands), and SWIR (remaining bands from 1100–2500 nm). The genetic algorithm conducts a search procedure based on an initial number (i.e., population) of combinations of features (i.e., individuals). The search procedure aims to maximize an internal optimization function (Figure 3, *desirability*, Kuhn [35]) that conjugates both an error metric (i.e., RMSE) and number of features. A response surface for the internal maximization (objective) function is presented in Figure 3. For each spectral range, the function's parameters (i.e., number of bands and RMSE) were set in a way that its range spanned across possible solution spaces for both initial and final generations. These ranges were found through an unconstrained model (*pls::plsr*), providing insight for achievable RMSE values. Otherwise, the GA could possibly be constrained in solution spaces where changes in performance would not necessarily equate to improvements in the objective function.

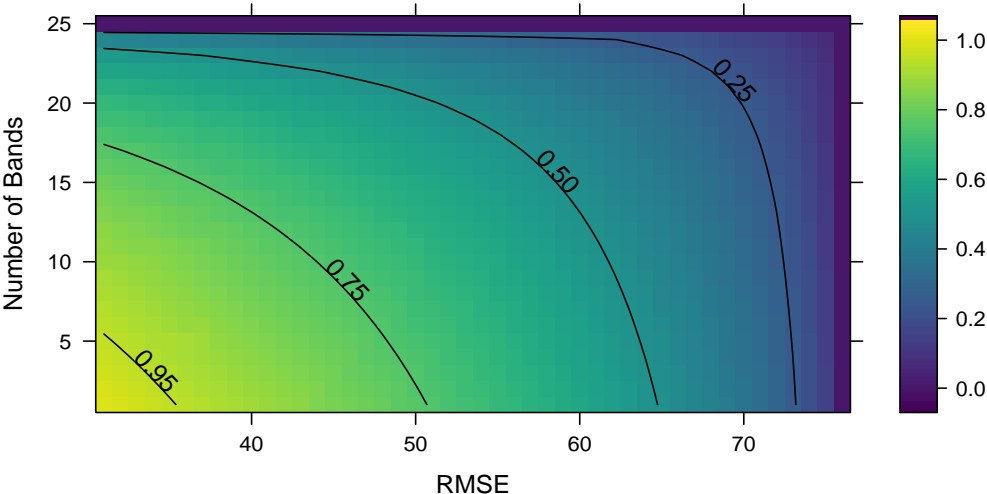

**Figure 3.** Internal maximization function surface and contour plot (*package::desirability*). The color gradient and contour lines indicate the response function of numbers of features and RMSE of both models. While RMSE has a linear component, the number of bands (features) was set to a quadratic function.

The best subsets of the feature space were found through the crossover of populations, mutation of individuals, and elitism, as described in in Scrucca [44]. A maximum number of generations (i.e., iterations) was set to 150 and the population size was set to 50. The range of 150 generations was set to provide an exhaustive search and, consequently, avoid either local minima or an early interruption of the search process. The population value of 50 was set to ensure a large population, in which the algorithm could best randomly select different subsets of bands [44]. To reduce the computational load, the maximum number of latent components was set to eight during feature selection. This number of latent variables was chosen so at least 95%–98% of the variance of the X matrix was utilized, ensuring comparison between models.

Each generation evaluates the overall fitness and selects the best individuals to be incorporated (as parents) into the next selection round (Figure 4). To assess possible overfitting, at each generation, an external assessment (i.e., RMSE) of the best individual was also computed. This assessment is linked to a 10-fold cross-validation where, at each fold, 90% of the data were used to train and 10% were held out as an external assessment. Consequently, the GA routine coupled with the internal maximization function was carried out 10 times for each spectral range and independent variable (i.e., either $CP_m$ or %CP).

A final selection procedure using all the 10-fold cross-validation's best individuals as the initial population was carried out against the complete data set. The individual with minimal RMSE is considered the optimal feature subset. In total, for each of the feature selection processes, the number of PLSR models (i.e., individuals) created was equal to 50 * 1500 as a result of the population size (50), the total number of generations (150), and the cross-validation process (10-fold cross-validation).

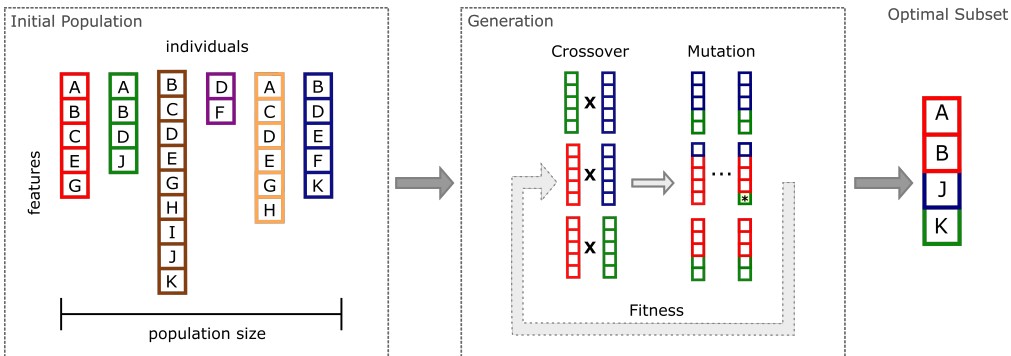

**Figure 4.** Genetic algorithm workflow. An initial set of candidate solutions is created from randomly selected features. These candidates are further selected based on internal fitness measurements. As a result, an optimal subset is identified. Tuning of the algorithm is done through adjustment of the population size, levels of crossover and mutation, number of generations, and fitness measurements.

Variable Importance: This was assessed using two different parameters: variable importance in projection (VIP; Equation (1)) and significance multivariate correlation with PLS (sMC; Equation (2)), which were calculated according to Mehmood et al. [36] and the associated R-package *plsVarSel*.

$$v_j = \sqrt{p \sum_{a=1}^{A} (SSY_{comp,a}) \times W_{a,j}^2) / SSY_{cum}} \tag{1}$$

$$sMC_j = \frac{MS_{j,PLS_{regression}}}{MS_{j,PLS_{residuals}}} \tag{2}$$

where:

$p$ :      number of variables;
$v_j$ :      variable importance in projection of the $j$th feature (band);
$a\ to\ A$ :   $a$th component to maximum ($A$) number of latent components;
$j$ :      the $j$th feature;
$SSY$ :      sum of squares of $Y$;
$W$ :      loading weights of the predictor matrix (i.e., features);
$MS$ :      mean squares (from the $F$-test notation).

VIP calculates the cumulative measure of the influence of individual features (variables) on the model. As a general rule, features with a VIP of lower than 1 are considered not important.

The sMC (Equation (2)) returns the $F$-test value of each predictor, while also calculating the $F$-value significance threshold (at $\alpha = 0.05$). Features with $F$-test values higher than the threshold level are considered not important.

C. Performance Assessment and Validation: Nested cross-validation was chosen to assess model tuning and performance. In it, hyperparameter optimization (i.e., number of latent components) was performed within the inner cross-validation (i.e., *bootstrap*), while the outer cross-validation computes an unbiased estimate of the expected model's average error and distribution. The outer validation, however, is performed using two different strategies: (I) a $k$-fold cross-validation (*rsample::v_foldcv*) and (II) a group $k$-fold cross-validation (*rsample::group_vfold_cv*). While the first was assigned to randomly

choose measurements from all experimental sites, the latter held out a single location at a time as the validation set. These two strategies should allow model comparison performance at locations with prior information and at unseen locations (Figure 5). The first strategy (Figure 5A.1,A.2) corresponds to the performance of the model when trained, tested, and validated using all locations (i.e., prior information). In the second strategy (Figure 5B.1,B.2), the validation is a particular held-out location (unseen locations).

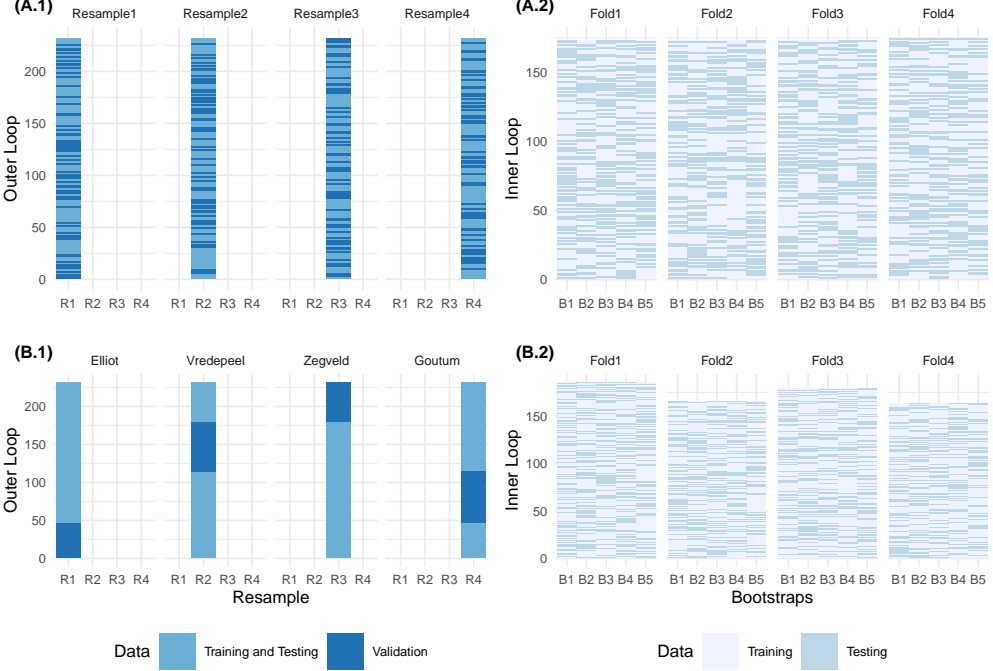

**Figure 5.** Validation strategies. **A.1,A.2**: correspond to a model validation strategy with (a priori) knowledge, respectively presenting the model performance assessment (**A.1**) and hyperparameter tuning (**A.2**). **B.1,B.2**: these correspond to model performance assessment in an unseen location. In both (**A** and **B**), hyper-parameters are tuned using bootstraps (B1 to B5 or the Inner Loop) and validated against different resample folds (R1 to R4 or the Outer Loop).

## 3. Results

### 3.1. Ground Truth Analysis

As presented in Table 2, the experimental design was appropriate for generating a large number of samples with a diverse range of biophysical and biochemical properties. The number ($n$) of observations acquired across all locations was equal to 266 (Elliot: 50, Goutum: 72, Vredepeel: 72, and Zegveld: 72). Following the removal of high biomass observations (i.e., sample weight above 3500 kg DM.ha$^{-1}$) and the outlier detection protocol, the final number of samples used for model development was reduced to 231. Most of the excluded data points (considered to be outliers or high-leverage data points) belonged to Zegveld's data set (20; Table 2). On closer inspection (i.e., digital photography), these detected outliers had a mixed botanical composition (white clover or *Trifolium repens* and perennial ryegrass), thus justifying their identification as outliers and removal of the data sets.

All locations displayed similar ranges for both %CP and biomass (Table 2), including both low and high values of %CP (e.g., between 10% and 27% %CP) and biomass (i.e., between 500 and 3000 kg DM.ha$^{-1}$). The sampling process at Elliot and Zegveld showed a uniform distribution across the CP$_m$ range, whereas the observations of Goutum and Vredepeel were skewed towards values below 200 kg CP.ha$^{-1}$ (Figure 6B).

Regarding comparability between locations, Bartlett's test (*p*-value < 0.01) rejected the $H_0$ (null hypothesis) under which $CP_m$ grouped by locations displays the same variance and, in context, was non-normally distributed. Given the non-normality, the KW test further indicated that there was a significant difference between distributions (*p*-value < 0.01). Finally, Dunn's test (using the Bonferroni adjustment method) indicated that while Vredepeel and Zegveld have different distributions, Elliot and Goutum were not significantly different (*α*-level = 0.05) from any other location (Figure 6A).

**Table 2.** Descriptive statistics of biophysical and biochemical attributes of data per location.

| Location | Samples (*n*) | | $CP_m$ (kg CP.ha$^{-1}$) | | | % CP (% DM) | | | Biomass (kg DM.ha$^{-1}$) | | |
|---|---|---|---|---|---|---|---|---|---|---|---|
| | Initial | Final | Mean | Min | Max | Mean | Min | Max | Mean | Min | Max |
| Elliot | 50 | 46 | 223 | 43 | 578 | 20 | 12.3 | 27 | 1115 | 224 | 2986 |
| Goutum | 72 | 68 | 208 | 76 | 601 | 16 | 9.4 | 28 | 1380 | 526 | 3282 |
| Vredepeel | 72 | 65 | 172 | 70 | 479 | 18 | 9.6 | 26 | 1015 | 395 | 3223 |
| Zegveld | 72 | 52 | 283 | 70 | 664 | 18 | 11.5 | 24 | 1566 | 592 | 3420 |

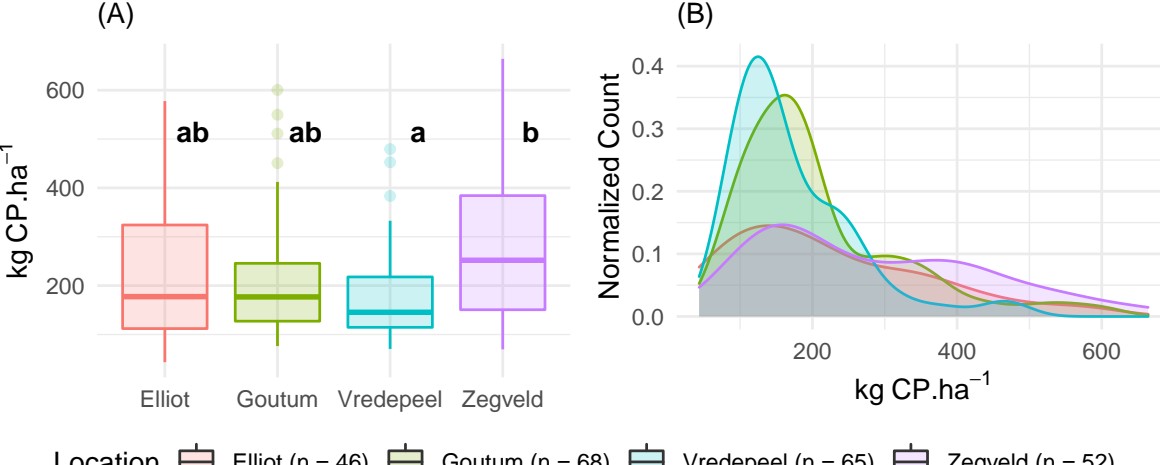

**Figure 6.** Boxplots (**A**) and density plots (**B**) of crude protein yield (kg CP.ha$^{-1}$) grouped by different locations. Dunn's multiple comparisons (*α*-level = 0.05) results are presented next to each boxplot. This numbers of observations (*n*) per location are shown in the bottom legend.

The Pearson correlation (*ρ*) between %CP and biomass was equal to −0.24 (*p*-value < 0.01), indicating that neither parameter was collinear, a possible factor for spurious regression models. As expected, the correlation between $CP_m$ and biomass was both closely related and significant (*ρ* = 0.91, *p*-value < 0.01). Lastly, the Pearson correlation between %CP and $CP_m$ was equal to 0.13 (*p*-value = 0.05), indicating that higher values of $CP_m$ were not strongly associated with increases in %CP and were, borderline, not significant.

### 3.2. Spectral Analysis

The diverse range of biophysical and biochemical properties generated a wide measurable range of spectral responses, which was further enhanced by the pre-processing protocol (i.e., Savitzky–Golay smoothing filter and removal of noisy spectral areas, Figure 2).

An exploratory analysis was performed by grouping and averaging spectral measurements within each $CP_m$ quartile (98, 151, 219, and 409 kg CP.ha$^{-1}$). The relationship between $CP_m$ and reflectance levels proved to be consistent throughout the spectrum (Figure 7a–c). Specifically, in two distinct ranges (from 400–750 nm and 1475–2500 nm), a negative relationship between reflectance and $CP_m$

was observed. Conversely, from 750–1325 nm, $CP_m$ and reflectance levels were positively related. On the contrary, signs of saturation between the two largest quartiles were evident in two spectral ranges (400–560 nm and 1475–2500 nm). The largest separation between spectra was located within the NIR portion (750–1325 nm) of the spectra (Figure 7c). Both data sets (i.e., VIS-NIR and SWIR) employed in the GA feature selection processes presented both direct and inverse relations between $CP_m$ and reflectance values. This allowed the regression algorithm to make use of these relationships without any explicit disadvantage when constraining the search space to either VIS-NIR or SWIR.

When employing the same exploratory analysis for %CP (quartiles: 12.9, 16.4, 19.1, and 22.7%), this separation was not clearly or linearly distinguishable (Figure 7d–f).

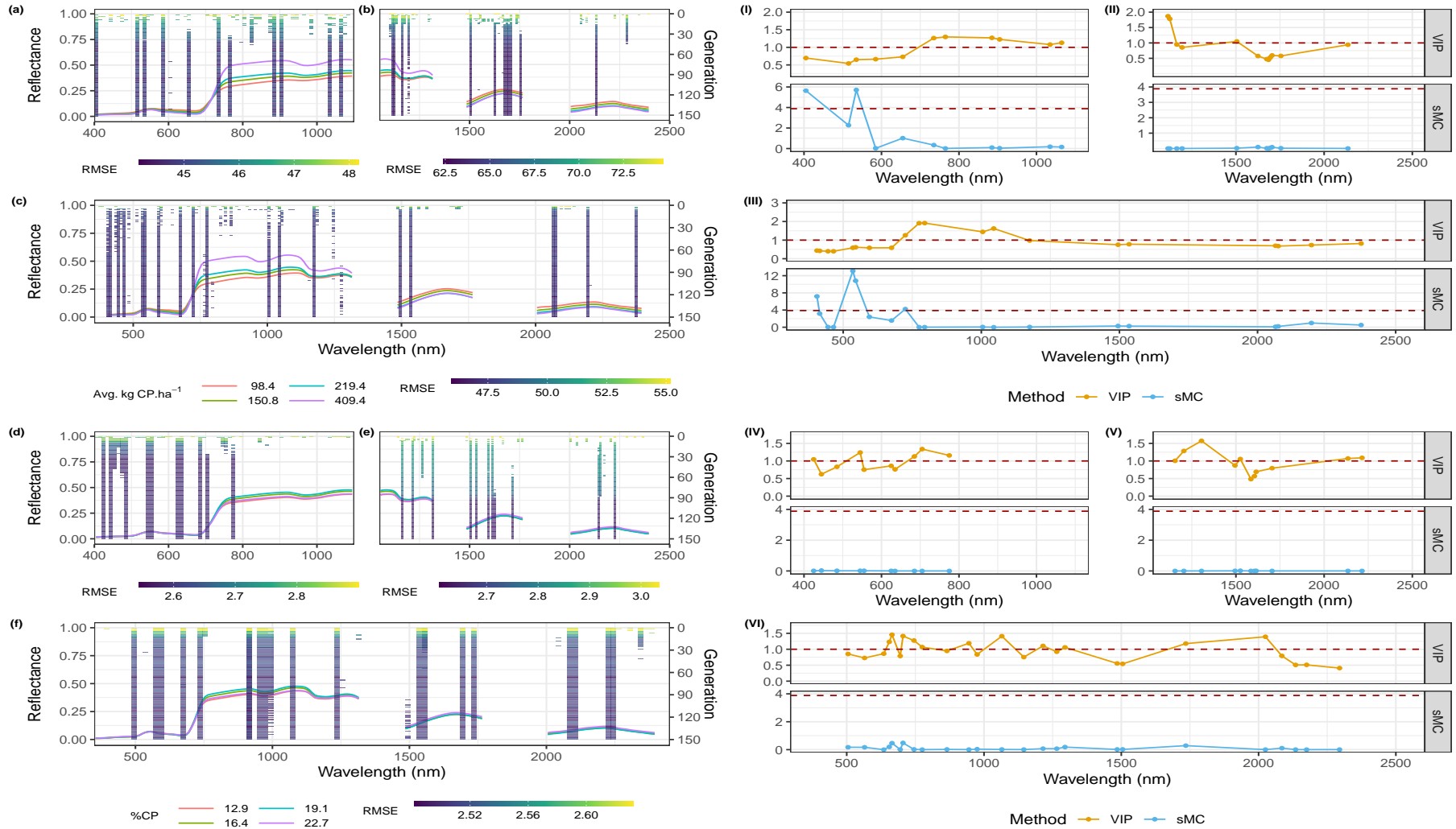

**Figure 7.** Feature (band) selection results of the genetic algorithm routine (left side) and corresponding variable importance proxies (right side). This upper side and lower side correspond to CP$_m$ (**a–c** and **I–III**) and %CP (**d–f** and **IV–VI**), respectively.

### 3.2.1. Feature Selection (Band) Selection

**Genetic Algorithm**: Given the massive computational load, at most folds and spectral ranges, the algorithm was able to narrow down the optimal number of features within approximately the first 50 generations (Figure 8). In most folds, similarity (as measured by the Jaccard Similarity, Jaccard [45]) between optimal individuals reached 90% in the 50th generation. However, similarity between folds was not an issue, and a comprehensive pool of features was selected for the final GA run.

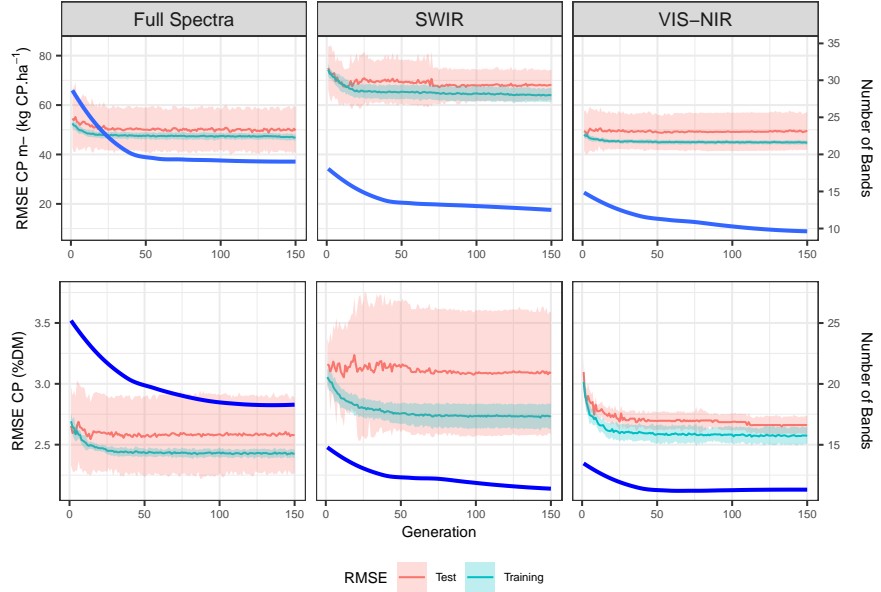

**Figure 8.** Performance during the genetic algorithm process. This RMSE average and confidence interval ($\mu \pm 1\,\sigma$) are shown on the primary axis. A locally estimated scatterplot smoothing (LOESS) of the number of bands (blue-color line) is presented in the secondary axis. Spectral ranges are presented in a column-wise fashion. This RMSE for $CP_m$ and %CP is presented in a row-wise fashion.

Additionally, given the extensive sampling process of the training (i.e., $n = 1500$) and test (i.e., $n = 150$) sets—and assuming the normality of residuals—95% of residuals should fall within the average RMSE $\pm 1.96$ standard deviation ($\mu \pm 1.96\sigma$), allowing one to infer whether the prediction accuracies within and between spectral ranges were different (Welch $t$-test, *stats::t.test*).

Lastly, the initial numbers of bands available at the start of each GA process were equal to 70, 91, and 161 for VIS-NIR, SWIR, and the full spectrum, respectively. In all processes, the algorithm was successful in decreasing either the number of features, the error metric, or both. To avoid excessive repetition and long numeric sequences, selected bands are provided as a supplementary material, although they are illustrated in Figure 7.

VIS-NIR Feature Selection Protocol

*Crude Protein Yield* ($CP_m$): The confidence interval ($\mu_{training} \pm 1.96\,\sigma$) for the training set was equal to $45.1 \pm 1.6$ kg CP.ha$^{-1}$ (Table 3). Within the test set, the confidence interval ($\mu_{testing}$) lies within $49.3 \pm 14.1$ kg CP.ha$^{-1}$, while results ranged from 35 to 63 kg CP.ha$^{-1}$. The average RMSE values of the test and training sets were statistically different (as per a two-sided $t$-test, $p$-value $< 0.01$), which may signal over-fitting. However, the true differences ($\alpha - level = 0.01$) between averages are expected to be within $-4.7$ and $-3.7$ kg CP.ha$^{-1}$, a small difference. As the test set results present a wider standard deviation (Figure 8), these results may indicate a difference in precision rather than accuracy.

In comparison, the RMSE values (test set) of the initial and final eleven generations are almost identical ($\mu_{initial} = 49.5$ and $\mu_{final} = 49.4$ kg CP.ha$^{-1}$). This difference was not statistically different

(*p*-value = 0.89); however, the average number of features decreased from 14.8 to 9.7, which indicates that the GA process was able to effectively reduce the number of features while keeping an equivalent level of accuracy and precision (Figure 8).

**Table 3.** Summary of the genetic algorithm routine's results for crude protein yield ($CP_m$), modeled using different spectral ranges.

| | Training (RMSE-kg CP.ha$^{-1}$) | | | | | | Testing (RMSE-kg CP.ha$^{-1}$) | | | | | | Features (*n*) | | |
|---|---|---|---|---|---|---|---|---|---|---|---|---|---|---|---|
| | $\mu$ | Max | Min | $\sigma$ | Ini | Fin | $\mu$ | Max | Min | $\sigma$ | Ini | Fin | $\mu$ | Max | Min |
| VIS NIR | 45 | 50 | 43 | 0.97 | 47 | 45 | 49 | 63 | 35 | 7.2 | 50 | 49 | 11 | 24 | 8 |
| SWIR | 65 | 79 | 60 | 3.23 | 71 | 64 | 69 | 93 | 56 | 7.6 | 71 | 68 | 14 | 33 | 9 |
| Full | 48 | 55 | 45 | 1.54 | 50 | 47 | 50 | 80 | 37 | 8.7 | 53 | 50 | 20 | 52 | 13 |

*Note:* Confidence intervals ($\alpha$ = 0.95) are expected to be within the mean $\pm$ 1.96 standard deviation ($\mu \pm 1.96\sigma$).

*Crude Protein As Dry-Matter Fraction* (%CP): The confidence interval ($\mu_{testing}$) was equal to 2.7 $\pm$ 0.7 %CP, while results ranged from 1.98 and 3.55 %CP (Table 4). When examining the performance of individuals of the initial and final eleven generations, a two-sided *t*-test indicates that there was a significant improvement in performance (*p*-value < 0.01). The performance improvement is approximately equal to 7.1% ($\mu_{initial}$ = 2.87% and $\mu_{final}$ = 2.66%), while there is a simultaneous decrease in the number of features (13 to 11, respectively) (Table 4).

**Table 4.** Summary of the genetic algorithm routine's results for crude protein as a dry matter fraction (%CP), modeled using different spectral ranges.

| | Training (RMSE-%DM) | | | | | | Testing (RMSE-%DM) | | | | | | Features (*n*) | | |
|---|---|---|---|---|---|---|---|---|---|---|---|---|---|---|---|
| | $\mu$ | Max | Min | $\sigma$ | Ini | Fin | $\mu$ | Max | Min | $\sigma$ | Ini | Fin | $\mu$ | Max | Min |
| VIS NIR | 2.6 | 3.2 | 2.4 | 0.09 | 2.8 | 2.6 | 2.7 | 3.5 | 2.0 | 0.38 | 2.9 | 2.7 | 12 | 21 | 8 |
| SWIR | 2.8 | 3.2 | 2.6 | 0.11 | 3.0 | 2.7 | 3.1 | 4.2 | 2.0 | 0.48 | 3.1 | 3.1 | 12 | 27 | 6 |
| Full | 2.4 | 2.8 | 2.3 | 0.06 | 2.6 | 2.4 | 2.6 | 3.3 | 1.9 | 0.31 | 2.6 | 2.6 | 20 | 41 | 12 |

*Note:* Confidence intervals ($\alpha$ = 0.95) are expected to be within the mean $\pm$ 1.96 standard deviation ($\mu \pm 1.96\sigma$).

In summary, the VIS-NIR spectral range presented both the best accuracy (smallest mean RMSE) and highest precision (narrowest standard deviation) for $CP_m$, while also requiring the smallest number of bands. For %CP, as per Table 4, it ranked second both in accuracy and precision while requiring the minimal number of features. It should be noted that this higher level of RMSE was due to the average RMSE of the initial generations, which were significantly different from the final generations (two-sided *t*-test *p*-value < 0.01).

In contrast, throughout the selection process, the VIS-NIR was consistently the spectral range with the highest precision (narrowest standard deviation), although its accuracy (average RMSE) was similar to that of the full spectrum (Figure 8). If the first 25 generations are set aside, the performance confidence interval for the VIS-NIR range was equal to 2.58 $\pm$ 0.13 %CP, best portraying its overall performance (Figure 8).

SWIR Feature Selection Protocol

*Crude Protein Yield* ($CP_m$): The average error for the training set ($\mu_{training}$) was equal to 65.3 $\pm$ 6.3 kg CP.ha$^{-1}$, presenting a range of 60 to 78.7 kg CP.ha$^{-1}$, the largest of the three spectral ranges. Accordingly, the test set presented the highest and widest RMSE values ($\mu_{testing}$ = 68.8 $\pm$ 14.9 kg CP.ha$^{-1}$), with results in the range of 55.5 to 92.6 kg CP.ha$^{-1}$ (Figure 8).

A two-sided *t*-test indicated that the initial and final eleven test sets ($\mu_{initial}$ = 71 and $\mu_{final}$ = 68 kg CP.ha$^{-1}$) were significantly different (*p*-value < 0.01), indicating that the GA routine was

able to simultaneously decrease the average RMSE and number of bands (14.7 to 11.4, respectively). The initial number of features included in the pool for the final GA run was equal to 47 out of 91 features, providing an ample subset for selection. The final number of features in the optimal individual was 11.

*Crude Protein as Dry-Matter Fraction* (%CP): The confidence interval of the training set ($\mu_{training}$) was equal to 2.8 ± 0.2, while the test set ($\mu_{testing}$) presented a higher and wider level: 3.1 ± 0.9 %CP (Figure 8). The final GA run pre-selected 34 of the 91 available features. During all runs, the minimum and maximum number of features selected within all generations ranged between 6 and 27 bands, a range narrower than only the full spectrum range (Table 4). Of all three spectral ranges, SWIR performed the worst, presenting the lowest accuracy and precision for both $CP_m$ and %CP.

Full Spectrum Selection Protocol

*Crude Protein Yield* ($CP_m$): The minimal RMSE (test set) was equal to 48.8 kg CP.ha$^{-1}$ and reached at the 132th generation. The initial generation of the GA process selected, at each fold, an average of 32 bands (Figure 8), reaching the minimal number of variables (i.e., 13) and maximum (i.e., 52) at the 149th and 1st generation, respectively. The final run of the selection uses 69 out of 161 features available and selected 19 bands as features for the optimal subset.

The range of RMSE found within the test set was 44.8 and 55.3 kg CP.ha$^{-1}$, with an average ($\mu_{training}$) of 47.5 and standard deviation ($\sigma$) of 1.3 kg CP.ha$^{-1}$. Consequently, the confidence interval (95% range or $\mu_{training} \pm 1.96\sigma$) of observations was within 47.5 ± 2.6 kg CP.ha$^{-1}$. In addition, for the test data set ($\mu_{test}$), the RMSE range (95% of the observations) was within 50.2 ± 16.6 kg CP.ha$^{-1}$). The averages (i.e., training and testing) can be compared using a two-sample *t*-test ($\alpha = 0.99$), by which the means were statistically different (*p*-value < 0.05), although the true differences between means were rather small (0.7 and 6.1 kg DM.ha$^{-1}$ at $\alpha = 0.05$).

When comparing the RMSE (test set) values of the initial generations (with more than 13 bands, $\mu_{initial} = 53.3$ kg DM.ha$^{-1}$) and those of the final generations (with 10 or fewer bands, $\mu_{final} = 49.9$ kg CP.ha$^{-1}$), the RMSE difference was statistically different (*p*-value = 0.01), and the true difference should lie ($\alpha = 0.05$) between 0.69 and 6.05 kg CP.ha$^{-1}$.

*Crude Protein as a Dry Matter Fraction* (%CP): Models employing the full spectrum have consistently presented the largest number of bands. Initially, the optimal individual had, on average, 23 features. These individuals ranged from 12 and 41 features, while the smallest test RMSE was found using 25, although models with less than 20 features had equivalent performance (Figure 8). This training and test ranges were equal to 2.3% and 2.8% as well as 1.9 and 3.3 %CP, respectively. For %CP estimations, on average, the FS range presented the smallest average RMSE of all spectral ranges; however, its confidence interval was wider than the VIS-NIR range (Figure 8). This confidence interval ($\mu_{testing} \pm 1.96\sigma$) for testing was within 2.6 ± 0.6 %CP (Table 4).

3.2.2. Variable Importance

The final feature sets selected by the GA process were also examined through variable importance proxies. The metrics for these proxies (Equations (1) and (2)) were considerably more elaborate than the internal maximization function used by the genetic algorithm. Consequently, these methods were able to further indicate candidates for feature selection, as presented in Figure 7 (I–VI—right-side). Each spectral range was evaluated by both the VIP (top) and sMC methods (bottom). The sMC thresholds (3.88) were the same across all models, given that sample sizes and, consequently, degrees of freedom were the same. For VIP, all features below the threshold of 1 were considered unimportant.

*Crude Protein Yield* ($CP_m$): Within the VIS-NIR spectrum (Figure 7I), the VIP method indicated that features below 700 nm were not important. The sMC indicated that both 405 and 535 nm were not

necessary for good model performance. This performance on the validation set (Figure 9), provides further evidence that the variable selection has successfully selected important variables.

For the SWIR range (Figure 7II), when the less important variables were within the 1600–1750 nm range, the most important bands were located in the NIR plateau (up to 1250 nm). However, throughout the bands of this spectral range, almost no differentiation between each spectrum was found (Figure 7e).

In the *full spectrum* model, the features centered at 725, 775, 795, 1005, and 1045 nm were considered the most important according to the VIP method (Figure 7III). Like in the VIS-NIR range, the bands within the visible range 405, 535, 545, and 725 were identified as having less impact on the model's overall performance.

*Crude Protein as a Dry Matter Fraction* (%DM): Overall, the models for %CP perform poorly (Figure 8). Consequently, variable selection should be examined carefully or disregarded. As seen in Equation (2), this proxy (i.e., sMC) measures the difference in explained variance (already quite low in poor-performing models). Consequently, its interpretation was of limited value.

Regarding the %CP of VIS-NIR, the features considered to be important were 425, 545, 685, 705, and 775 nm. In contrast to $CP_m$, all selected variables were mostly in the visible range and below 800 nm.

For the SWIR, no clear patterns can be identified, as the VIP method selects 1155, 1205, 1305, 1525, 2135, and 2215 nm, while the sMC would not discard or indicate a strong candidate for elimination.

Within the full spectrum only 11 out of 25 were considered important. Given that the VIP method is a translation of each feature weight across latent variables (Equation (1)), the pattern exhibited in Figure 7VI suggests that the model has been over-fitted, where contiguous variables present alternating high weights. This condition (i.e., overfitting) was further evidenced by the model performance analysis (Figure 10II,VI).

### 3.2.3. Performance Assessment

*Crude Protein Yield* ($CP_m$) VIS-NIR: The validation results for the models trained with prior knowledge were equal to 79.2 kg CP.ha$^{-1}$ (Figure 9I). The models presented a consistent behavior when validated in unseen locations, displaying errors equal to 91.6, 88.3, 90.5, and 73.8 kg CP.ha$^{-1}$ for Elliot, Zegveld, Vredepeel, and Goutum, respectively. Overall, the error across sites was equal to 85.5 kg CP.ha$^{-1}$ (Figure 9IV).

For both *SWIR* and *Full Spectrum*, a large unequal residual dispersion was present up to 300–400 kg CP.ha$^{-1}$ (Figure 9—SWIR and Full Spectrum). In contrast, the models based on the VIS-NIR present a higher level of homoscedasticity. Although the full spectrum presents a smaller RMSE for known locations, this may be seen as a random artifact of the seed value used for data partitioning given that both present the same average and standard deviation within testing data sets (Table 3).

As discussed, the VIP indicates a possible overfitting for models based on the full spectrum. Given this, the results for a more thorough testing (i.e., unseen locations) show that the algorithm was not able to keep the same level of accuracy. Its inaccuracies have a two-fold increase across validation strategies Figure 9III,VI).

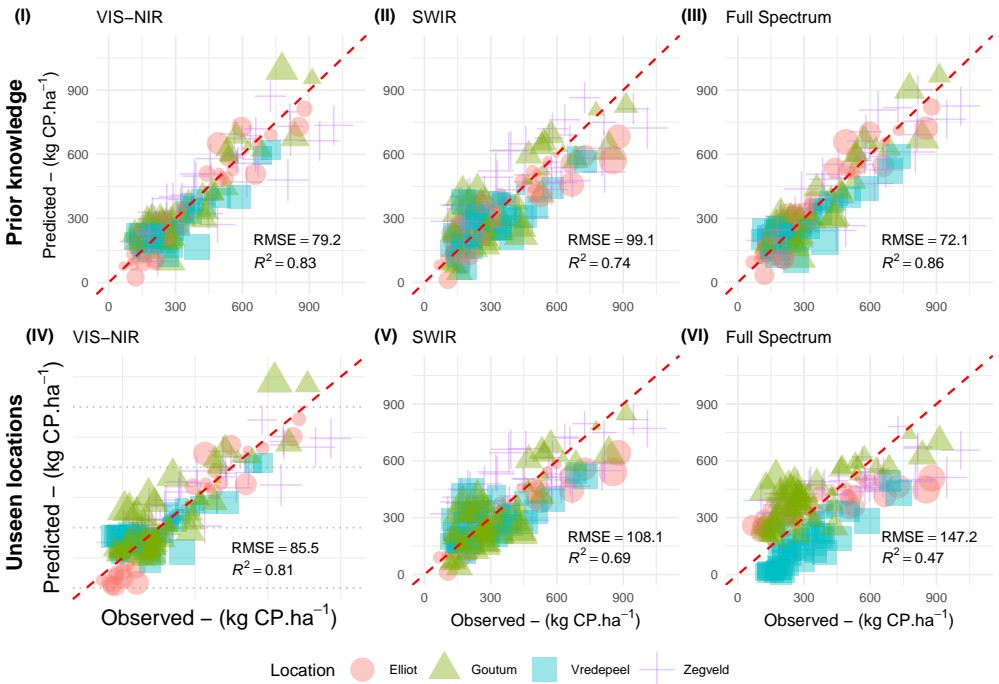

**Figure 9.** Scatterplots *Predicted* by *Observed* using the bands selected through the GA band selection routine for $CP_m$ of each spectral range (column). This top row corresponds to the partial least squares regression (PLSR) algorithm trained in all locations. This bottom row corresponds to PLSR algorithms trained while withholding the test location.

*Crude Protein as a Dry Matter Fraction* (%CP): All models display a low level (i.e., $R^2 < 0.5$) whenever assessed against unseen locations. In addition, it is possible to notice that there is a significant clustering of predictions per location within Figure 10V,VI). Between spectral ranges, for models trained with prior knowledge, the results range from RMSE 2.6 to 2.7 %CP. For unseen locations, performances decreased to a range between 3.1 to 3.5 %CP (Figure 10). Similarly to $CP_m$, performances decreased when validated for unseen locations. From the three ranges, the VIS-NIR provided the smallest decrease in performance when assessed against unseen locations. The full spectrum and SWIR had their coefficient of determination more than halved (Figure 10—SWIR and full spectrum).

For all ranges, values above 25 %CP were poorly predicted.

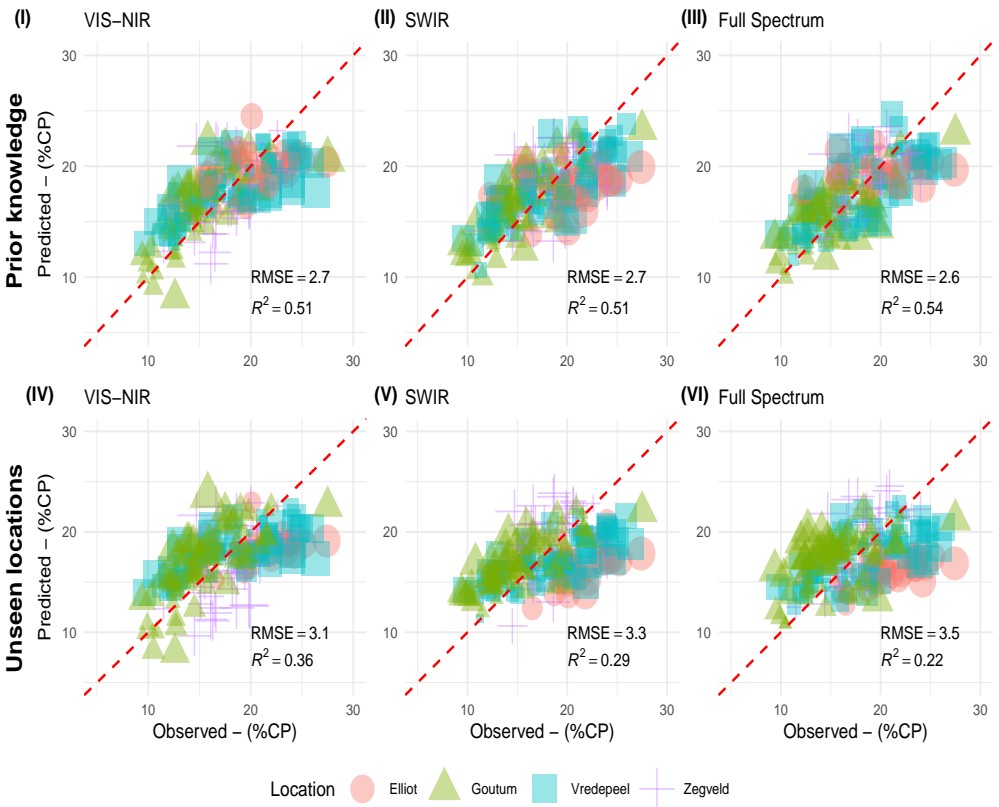

**Figure 10.** Scatterplots *Predicted* by *Observed* using the bands selected through the GA band selection routine for %CP of each spectral range (column). This top row corresponds to the PLSR algorithm trained in all locations. This bottom row corresponds to the PLSR algorithms trained while withholding the test location.

## 4. Discussion

This study aims to establish an effective and affordable approach for a non-destructive and near-real-time crude protein retrieval based solely on top-of-canopy reflectance. It has successfully achieved its aims: Firstly, the results showed that crude protein can be best explained, in field conditions, as a function of mass per area ($CP_m$) rather than as a fraction of biomass (%CP). Secondly, crude protein can be best estimated using the visible to near-infrared range of the spectrum (400–1100 nm), a range associated with low-cost Si-based sensors, instead of the shortwave infrared range (1100–2500 nm), as also indicated in Shorten et al. [15] and Starks and Brown [20]. Thirdly, only a small number of bands (approximately eleven) are necessary for adequate model development. Despite a performance reduction, these (multispectral) models can be satisfactorily employed in unseen locations. Finally, these results indicate that a reasonably low-cost multispectral camera (or sensor) could—regardless of location—estimate perennial ryegrass $CP_m$ with an average error (i.e., RMSE) of close to 85 kg CP.ha $^{-1}$ within biomass levels of up to 3500 kg DM.ha $^{-1}$ (Figure 9IV). It is reasonable to assume that these models could be further refined to a level close to 80 kg CP.ha$^{-1}$ RMSE (Figure 9I), provided that models are updated for new locations, similarly to NIRS laboratory methods.

This study has found results equivalent to those of Kawamura et al. [19], despite employing a fraction of the number of bands proposed in that study. Kawamura et al. [19] selected 167 narrow bands (FWHM = 1 nm) for $CP_m$ estimation, while this study indicates that eleven broad bands (FWHM = 10 nm) have an equivalent or superior performance (Figure 8). Our results are also in agreement with previous studies [46], both when indicating the best-predicting spectral range (400–1100 nm) and the comparable prediction accuracy. Equally as important, for in-field conditions, the SWIR region did not consistently improve protein estimations, either when expressed as $CP_m$ or as

%CP (Figure 8—SWIR and FS), which does not justify its use for CP estimation. These results extend the findings of Kattenborn et al. [25], where the authors explicitly state that chlorophyll could be best estimated in its $\mu g$.cm$^{-2}$ form rather than on a %DM basis.

Our results showed that %CP prediction error is within the range of 2.6%–2.7% (Figure 10—"Prior knowledge" row) for observations at known conditions, while high levels of %CP (i.e., above 22.7 %) are poorly predicted (Figure 10). In more rigorous assessments (i.e., unknown locations), inaccuracies are above 3 %CP. In contrast to Kawamura et al. [19] and our results, Starks et al. [24] report a higher achievable accuracy (i.e., around 1.5 %CP). That study was performed, however, for a different grass species (i.e., bermuda grass—*Cynodon dactylon*), under a smaller biochemical range (i.e., 4–14 %CP), and for a consistent canopy architecture (i.e., bermuda grass's leaves are oriented parallel to the ground level—planophile).

In our regression models, most of the residuals are within the largest %CP quartile (above 22.7 %CP). Similarly to Starks et al. [24], the range below 15 %CP is well explained (Figure 10). Paradoxically, in well-managed pastures, perennial ryegrass's crude protein levels rarely drop below 18–20 %CP. However, a good spectral separability can be found between the three first quartiles of observations (12.9, 16.4, and 19.1 %CP), allowing for qualitative grading as per the Australian Fodder Industry Association (AFIA-Australian Fodder Industry Association Standards [47]).

As reported in Kawamura et al. [19], the feature selection routine (i.e., genetic algorithm) was shown to improve the model's final performance. In all cases, the procedure was able to simultaneously reduce the number of features and RMSE (Figure 8). In doing so, this routine highlighted important spectral regions, particularly the range between the red edge and the NIR shoulder (Figure 7a,I).

The range of 700–800 nm has been consistently reported when estimating biomass and nitrogen content [23,48], while being less prone to illumination effects [49]. Differently from other spectral regions, this region has a shift (i.e., the red-edge shift) towards wider wavelengths [50], consistently linked with higher chlorophyll concentration and LAI, important and correlated features for CP. This has been explored by Mutanga and Skidmore [48] to estimate %N content with high accuracy for Buffel grass (*Cenchus ciliaris*) in a greenhouse experiment. The authors [48] ([p. 41]) expressly indicate the spectral ambiguity that could arise from different %CP and biomass levels as a significant limitation in their assessments.

These findings warrant the parsimonious selection of features rather than the common practice of arbitrary employment of all sensors' outputs. Although many regression algorithms should cope with multicollinearity or irrelevant variables, our results illustrate how a parsimonious model (Figure 9I,IV) is more generalizable than a model with a higher number of features (Figure 9III,VI). These results also indicate the need for judicious planning of validation strategies. While both the VIS-NIR and FS have a similar performance in an unconstrained $k$-fold cross-validation, the VIS-NIR model significantly outperforms the FS when using unseen locations as the validation set.

When examining the overall pattern of meaningful features for CP$_m$ (Figure 7I), it can be argued that these are convergent with the area of highest separation between spectra, which was found in the initial explanatory analysis (i.e., 750–1325 nm). As discussed by Kjeldahl and Bro [26] ([p.559]), the use of relevant variables (i.e., spectral range) is a crucial step in the development of efficient models. In a coherent fashion, most variables indicated to be important by the VIP are within the 725–1325 nm range (Figure 7a–c). Unfortunately, the arbitrary pre-selection of a spectral range is not straightforward, as crude protein is present in a wide range of cellular constituents, and several overlapping absorption effects are present. Additionally, as shown in Figure 7I–VI, variable importance proxies are not always convergent, and further selection could be addressed through different techniques based on experts' discretion and trade-off evaluations.

Nitrogen is a highly mobile plant nutrient, present in many forms [51] and performing complex interactions within plants. Given these complex interactions in a non-controlled environment, fundamental spectroscopy methods—such as those employed in organic chemistry at the laboratory level—seem to be inappropriate in field conditions. Consequently, one could hypothesize that the

best strategy for practical purposes is to rely on correlation rather than causal effects. In other words, measurements are based not on molecular bonds (e.g., overtones), commonly associated with narrow band signals, but rather on the correlation of main drivers of absorption (broad band effects), such as plant pigments and cell walls. Our best methods—although efficient—may be mapping (biological) correlations and not causal (e.g., overtones) indicators of protein (in its diverse forms) [51].

Our results show that the ability to estimate %CP at the paddock level solely from TOC reflectance is questionable or, at least, inferior to $CP_m$ estimation (Figure 8). Fundamentally, Beer's law directly relates light absorption as the product of concentration and optical thickness. To disregard a proxy of an optical path (e.g., biomass) will necessarily result in an underdetermined (or poorly determined) system. This ambiguity has been discussed in depth by different authors [11,12,23,50], and their findings are either in agreement or provide a strong background to support our results and limitations.

Although our experimental design does not aim to clarify these issues, it is reasonable to suggest that the measurement of CP as mass per area indirectly incorporates effects that are linked to optical path (e.g, LAI). Other sources of ambiguity, such as leaf angle distribution, may also be alleviated by the inclusion of biomass per area, given that, despite not being accurately determined, this factor presents a distribution of values per biomass range. This underdetermination could be better approached through the use of radiative transfer models. However, given the complex nature of the necessary measurements, at this moment, these solutions seem inadequate for a farm scenario.

It should be stated that, in laboratory conditions, the seminal NIRS work of Marten et al. [32] (Table 5) employed only 5–7 bands for %CP estimation without any spectral transformation, achieving a high level of accuracy (RMSEP = 0.5% and $R^2$ = 0.99). In field conditions, however, moisture content and canopy geometry will necessarily overlap and mask biochemical spectral features. In addition, the main advantage of hyperspectral data does not reside in the high number of bands, as contiguous bands are highly correlated, but in its application to derive spectral features, such as the continuum-removed features (e.g., band depth, feature area), or to resolve overlapping signals [52,53]. However, although frequently and successfully employed in a laboratory setting, these techniques would require high radiometric accuracy in outdoor environments (and absence of the mentioned masking effects), rendering them unfeasible for precision agriculture purposes. Consequently, a custom-built multispectral system—such as those proposed in this study—should be the most appropriate solution for an agricultural scenario.

Limitations: It is important to highlight that $CP_m$ and DM ($\rho$ = 0.91) are highly linked; however, this should not be understood as a spurious relationship, given that biomass, LAI, and path length are components of TOC reflectance. However, these relationships should be analyzed well so the causal relationships between biochemical components can be separated from the biophysical effects. In essence, we advocate that %CP estimation (concentration) cannot be solved unless the TOC reflectance is corrected for biomass (path length) effects.

Another limitation of this study is the instrument's different point-spread functions for each of the three built-in sensors (VIS-NIR, SWIR 1, and SWIR 2 ), as discussed in Mac Arthur and Robinson [30]. When not employing an optical scrambler, the response function within the instrument footprint is different for the internal sensors. Although this is most likely negligible, our experimental setup has not consciously controlled for confounding factors or issues that could further enhance this problem (e.g., broken optical fibers or heterogeneity within the canopy). Imaging spectroscopy and higher spatial resolution could clarify whether differences of %CP within the footprint may offset our method and enhance our understanding of the natural variability within the canopy. As noted by Curran [11], the biochemical composition within a canopy is variable, and how to best model its spatial variability "requires the imaginative use of three-dimensional spatial statistics, giving the most weight to the uppermost layers in the canopy, rather than simple random sampling representation" [11]. Employing a hyperspectral imaging camera, Shorten et al. [15] were able to provide a visualization of this while reporting differences in %N (equivalent to %CP) throughout the canopy strata.

An additional limitation of the applicability of these findings, as well as a topic for future research, is the scaling issues introduced by higher-altitude data acquisition (i.e., low-level flight) and different spatial resolutions. Many of these issues have already been discussed by Wang et al. [54] and Burkart et al. [55], indicating that these can be overcome in an operational scenario. However, another important factor is the measurement error associated with multispectral imaging sensors (e.g., noise reduction, dark offset, vignette and row-gradient correction, and lens distortion), which is also associated with custom-made multi-camera 2D imaging systems and has not been included in our method (nor was it the aim of the study to do so). These limitations can be operationally resolved to a high level of radiometric accuracy, as demonstrated in Mamaghani and Salvaggio [56].

From a sensing perspective, another important limitation is given by the selection of two bands beyond the 1000 nm (Figure 7–I), concurring with the diminishing spectral sensitivity of Si after 1000 nm. This constraint is trivial, however, given that spectral behavior from 800–1100 nm is highly correlated with and driven by broad spectral features (e.g., cell-wall absorption feature).

## 5. Conclusions

This study has employed a modified feature selection technique based on a genetic algorithm approach to determine the best spectral region, as well as the optimal and minimal numbers of bands when estimating crude protein. Additionally, it has also contrasted how to best portray crude protein (either as a dry matter fraction or on a weight-per-area basis) when utilizing top-of-canopy reflectance measurements for its estimation. Correspondingly, these tasks aimed to instruct the design of a new sensor (either point measurement or an imaging system).

Our results indicate that an affordable spectral-based sensor could estimate perennial ryegrass crude protein in outdoor environments using only eleven broad bands (10 nm bandwidth) within the visible to near-infrared range (400–1100 nm), provided that the protein is expressed in weight per area. Additionally, the models are transferable to new locations with a small decrease in performance (from 80 to 85 kg $CP.ha^{-1}$ RMSE). These results could lead to the development of sensors for autonomous deployment in an unmanned aerial or ground vehicle, providing end-users with essential information for the best agronomic practices.

This study displays a rigorous comparison between ways of portraying biochemical properties and model transferability under a robust theoretical and analytical framework. Finally, in light of future applications and technology adoption, it is necessary to reevaluate how feed quality parameters are expressed whenever canopy reflectance measurement is the ideal technique for data collection. Differently from benchtop near-infrared analysis, for field conditions, biochemical parameters should be expressed as a conjugation of their concentration and optical path (e.g., weight per area, $kg.ha^{-1}$), rather than as they are in standard laboratory results (i.e., concentration, % dry matter).

To all effects, using estimates of crude protein as mass rather than a fraction of biomass does not limit the management of a herd's diet or fertilization inputs. By expressing crude protein on a weight-per-area basis, the method and instruments presented can be used as a monitoring tool. Future research should focus on the translation and validation of our findings to a multispectral imaging sensor. Finally, different feed quality parameters could employ the method proposed in this study to determine achievable prediction accuracies and a minimal number of bands necessary for their retrieval.

**Author Contributions:** All authors conceived and designed the experiments; G.T.D.A. and G.R. performed the experiments; G.T.D.A. analyzed the data; all authors contributed with materials/analysis tools; G.T.D.A., A.L., L.K. and R.R. wrote the paper. All authors have read and agreed to the published version of the manuscript.

**Funding:** This research was supported by Dairy Australia, through the Dairy on PAR action (Tasmania). This research on the three experimental farms in The Netherlands has been carried out in the context of the Public-Private Partnership Precision Agriculture 2.0, financed by the Ministry of Agriculture, Nature and Food Quality, Agrifirm Plant B.V., ZLTO and Kverneland Group Mechatronics B.V. and in the context of the Public-Private Partnership Amazing Grazing, financed by the Ministry of Agriculture, Nature and Food Quality and ZuivelNL.

**Acknowledgments:** The authors would like to acknowledge the in-kind contributions of Eurofins Agro NL, which provided the feed quality analysis for data collected in the Netherlands. In addition, the authors would like to express their gratitude to ASD Malvern PanAnalytical, which provided the FieldSpec 4 through the Goetz Instrument Program. Finally, we would like to thank Chris Maclellan for offering a deeper understanding of the instruments used in this project.

**Conflicts of Interest:** The authors declare no conflict of interest.

## Abbreviations

The following abbreviations are used in this manuscript:

| | |
|---|---|
| CP | Crude Protein |
| $CP_m$ | Crude Protein Yield |
| *%CP* | Crude Protein as a Dry Matter Fraction |
| DM | Dry Matter |
| FS | Full Spectrum |
| InGaAs | Indium Gallium Arsenide |
| LAI | Leaf Area Index |
| LAD | Leaf Angle Distribution |
| MS | Multispectral |
| N | Nitrogen |
| NIRS | Near Infrared Spectroscopy |
| PbS | Lead Sulfide |
| R | Reflectance |
| RMSE | Root Mean Square Error |
| RMSE | Root Mean Square Error Prediction |
| RS | Remote Sensing |
| Si | Silicon |
| SWIR | Shortwave Infrared |
| TOC | Top of Canopy |
| VIS–NIR | Visible and Near Infrared |

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
