# Peer review of "Retrieval of Crude Protein in Perennial Ryegrass Using Spectral Data at the Canopy Level"

_remotesensing, doi:10.3390/rs12182958_

Round 1

Reviewer 1 Report

Summary:

Research objective was to establish an effective and affordable approach for a non-destructive, real-time estimation of crude protein (CP) in perennial ryegrass, based on top of canopy reflectance. Feature selection based on a genetic algorithm successfully selected spectral band subsets within different regions and across the full spectral range, minimizing both number of bands and an error metric. Visible to near-infrared range (400 - 1100 nm) utilizing only eleven broad bands (10 nm bandwidth) found to be the most successful in predicting crude protein expressed in weight per area, while calibrations could be transferred to new locations with a small decrease in performance. These results could lead to the development of sensors in an unmanned aerial or ground vehicle, providing relevant information for best agronomic practices.

Broad comments:

Research falls in the scope of the journal. Hypothesis and aim are well defined, while results represent advance to current knowledge in the field of remote sensing in agriculture, but particularly in spectral data analysis and agronomic information retrieval by sophisticated extraction of spectral variables (bands) important for CP estimation.

Introduction paragraph is concise, understandable, progressively give insight to the topic supported by relevant literature review.

Study is correctly designed with exhaustive, in detail and step-by-step elaborated methodology. Statistical methods are very well explained. Methodology is the most important part of the research, explaining how to extract relevant information/CP variability from reflectance data.

This is about novel approach (GA represents high level data mining compared to correlation matrices or λ matrix), topic is very actual and still under intensive research worldwide. Therefore, each effort to extract information enough relevant for remote assessment of crop biochemical variables is welcome and worthwhile. However, results obtained from different types of spectral sensors and from various statistical approaches can vary due to spatial and spectral resolution, number and diversity of samples, type of crops, spectral bands, spectra preprocessing and the sophistication of the prediction models.

As authors have stated, RS for precision agriculture should be simplified and cost effective, but accurate enough for seasonal crop monitoring and to support VRT. Research is aimed in that direction.

Results are concise and presented in visually clear, attractive and understandable form, with the highest standards.

Discussion is argumentative and pragmatic.

English is appropriate.

Few questions for authors:

So, N fertilization levels were chosen to achieve preferred data variability in the datasets/locations/plots? I assume it was not important to differentiate between N levels based on reflectance?

Specific comments can be found embedded in the text of the manuscript (.pdf).

Reviewer 2 Report

Comments and suggestions: The authors evaluated different spectral ranges, i.e., VIS-NIR, SWIR, and full range spectrum, for assessment of crude protein, which is either expressed as a dry matter fraction or in a weight per area basis, using genetic algorithm and PLSR. Abundant work has been involved in the manuscript, and extensive detailed information on data analysis have been presented. Nevertheless, the novelty of the research seemed to be not enough, since the manuscript just selected several band features for crude protein using PLSR. These similar analysis have been presented in some previous researches, the authors should emphasize the difference of their work compared to previous studies. Moreover, I believe the manuscript should clarify several crucial issues which is presented in the general comments below, and some minor specific comments also should be addressed for the improvement of the manuscript. General comments 1. For the introduction section, ONLY two references (ref.14 and ref.19) that related to crude protein estimation with remote sensing were mentioned. this is too simple for an introduction part of a manuscript. the authors must add relevant references and summarize the existing problem from these references and thus conduct a research. 2. For the methods section, The authors mentioned three dates “date 1, date 2, and date 3” in Table 1 for field data collection. For different sites, the specific dates were different, particularly for Elliot site, the three dates were so close. The authors should explain the reasons for choosing these dates for field data collection. 3. For canopy spectrum measurement in line 163 – 183, the authors should add the time and weather conditions for ASD measurements. 4. For spectral resample in line 225-226, the authors must provide detailed information on resample method, and resampled bands. Since different resample methods have direct effect on resampled spectrum. 5. For the number of generations and population size in line 247-248, the authors should explain the reasons for setting 150 and 50 to these two parameters, respectively. 6. For Variable Importance assessment in line 263 – 265, the manuscript used two different parameters, VIP and sMC. The authors were intend to use both two methods for band features selection? If so, how to choose band features from these two methods, since there might exist differences between these two methods according to Figure 8I to 8VI. If not, why the authors used two approaches, the manuscript should make this clear to readers. Specific comments 1. In line 42, the full abbreviation of ‘DM’ in this line is suggested. 2. In line 90, ‘esimations’ should be ‘estimation’. 3. In line 464, ‘Figure 10-I’ should be ‘Figure 9-I’. 4. In line 469, The full abbreviation of ‘FS’ in this line is suggested. Since it is so confusing. 5. The order of the figures throughout the manuscript should be renumbered. For example, Figure 8 in line 325 appeared earlier than Figure 7 in line 340.

Reviewer 3 Report

This paper aim is to provide a non-destructive method to evaluate crude protein in Perennial Ryegrass – very significant for livestock grazing - after some processing and therefore contribute to have farmers make informed decisions about livestock feeding management. While there are other studies that address this issue, the authors intended their method to be carried out in-situ,- by means of one remote sensing platform - preferably by means of a sensor developed to this purpose, based on their work. Moreover, this sensor will focus only in the spectral bands deemed more suitable for this task, as determined in this study. This contribution can be a step onward in providing farmers with management tools that can have a real impact in more sustainable agricultural practices without compromising yield or, in this case, livestock development and well-being.

As a general appreciation, the paper is very well written, is easy to grasp and to comprehend. I did not detect misspells or phrases that I would like to see changed. Furthermore, it is well structured and has all the components that I consider necessary to present a proper study of this sort.

Whilst the summary is very good, introduction’s first paragraph needs to have some scientific support, as some important considerations are made but not fully justified. Other than that, it nicely structured and presented. Regarding table 1, why was data acquired during November in Elliot? It falls out of pattern when compared with the remainder data acquisition processes and was not explained. As I am not aware of the growth season in Australia is it related? In Europe I understand the patterns so please fully explain it for all of us readers. Section 3 is a little longer and denser than needed but it provides the level of detail that can give a reader the opportunity to fully understand not only what was done but how it was planned.

I do not have anymore suggestions. As stated, this work is very nicely done, justified and supported. It does not provide ground breaking contribution to the state of the art, but the results – namely those 11 bands – can be very important to future work to be done in precision agriculture.

Round 2

Reviewer 2 Report

Thanks to the authors for considering the comments and suggestions. In the present version of the manuscript, the authors have explained and answered the general comments accordingly. Nevertheless, some specific comments listed below, which were mentioned previously, have not been tackled.

in line 42, the full abbreviation of ‘DM’ in this line is suggested.

in line 90, ‘esimations’ should be ‘estimation’.

in line 469, The full abbreviation of ‘FS’ in this line is suggested.

These minor problems should be addressed throughout the whole manuscript before acceptance.

Author Response

Dear Reviewer 2,

Thank you for your contribution and please excuse these lapses. The following changes were made to the manuscript:

- Regarding line 42 (first manuscript). Following the journal's instruction to authors:

"The abstract must be self-contained. A reader should not have to read through the paper to understand it. Therefore, please define any acronyms."

As suggested we have replaced the acronym "%DM" by "percentage dry matter"; thus, avoiding the use of acronyms in the Abstract section.

- Line 90. As suggested, the typo "esimations" was corrected.

- Line 469. As suggested, the original sentence "Although the FS presents a smaller RMSE for known locations,..." was altered to ""Although the Full Spectrum presents a smaller RMSE for known locations,..". 
We have also indicated  the definition of the acronym FS (full spectrum) when it is first used (i.e., "Introduction" section of the new manuscript).

To avoid an unnecessary round of revision, if these changes are not what you have requested, please let us know (possibly through the "Assistant Editor"), and the changes will be made during the author's proof-reading stage. 

Finally, we would like to thank you again for your review, comments and critiques. 

Kind regards, 
The authors